# Imaging Approaches to Investigate Pathophysiological Mechanisms of Brain Disease in Zebrafish

**DOI:** 10.3390/ijms24129833

**Published:** 2023-06-07

**Authors:** Lapo Turrini, Lorenzo Roschi, Giuseppe de Vito, Francesco Saverio Pavone, Francesco Vanzi

**Affiliations:** 1European Laboratory for Non-Linear Spectroscopy, Via Nello Carrara 1, 50019 Sesto Fiorentino, Italy; turrini@lens.unifi.it (L.T.); roschi@lens.unifi.it (L.R.); devito@lens.unifi.it (G.d.V.); francesco.pavone@unifi.it (F.S.P.); 2Department of Neuroscience, Psychology, Drug Research and Child Health, University of Florence, Viale Gaetano Pieraccini 6, 50139 Florence, Italy; 3Interdepartmental Centre for the Study of Complex Dynamics, University of Florence, Via Giovanni Sansone 1, 50019 Sesto Fiorentino, Italy; 4Department of Physics and Astronomy, University of Florence, Via Giovanni Sansone 1, 50019 Sesto Fiorentino, Italy; 5National Institute of Optics, National Research Council, Via Nello Carrara 1, 50019 Sesto Fiorentino, Italy; 6Department of Biology, University of Florence, Via Madonna del Piano 6, 50019 Sesto Fiorentino, Italy

**Keywords:** zebrafish, functional imaging, behavior, neurological disorders, brain disease models, epilepsy, Alzheimer’s, Parkinson’s, autism spectrum disorders, myelination

## Abstract

Zebrafish has become an essential model organism in modern biomedical research. Owing to its distinctive features and high grade of genomic homology with humans, it is increasingly employed to model diverse neurological disorders, both through genetic and pharmacological intervention. The use of this vertebrate model has recently enhanced research efforts, both in the optical technology and in the bioengineering fields, aiming at developing novel tools for high spatiotemporal resolution imaging. Indeed, the ever-increasing use of imaging methods, often combined with fluorescent reporters or tags, enable a unique chance for translational neuroscience research at different levels, ranging from behavior (whole-organism) to functional aspects (whole-brain) and down to structural features (cellular and subcellular). In this work, we present a review of the imaging approaches employed to investigate pathophysiological mechanisms underlying functional, structural, and behavioral alterations of human neurological diseases modeled in zebrafish.

## 1. Introduction

Neurological disorders nowadays represent the leading cause of disability and the second cause of mortality worldwide [1]. As a result of population growth and aging, in the coming decades, the number of people suffering from pathologies affecting the nervous system will substantially increase, making research on neurological disorders a social priority. In this context, the identification of the cellular and molecular pathophysiological processes underlying the onset and progression of neurological diseases represents a fundamental goal of research efforts aimed ultimately at the discovery of effective treatments.

In this framework, zebrafish (*Danio rerio*) has recently emerged as an extremely valuable animal model. Besides its well-known features such as external fertilization, small dimensions, tissue transparency, short generation time and reduced rearing costs that made this organism so appealing to researchers in the most diverse application fields, zebrafish exhibit a few other interesting properties that provide unique advantages for basic and translational neuroscience research. Even though evolutionary history of humans and fish diverged about 450 million years ago [2], as a vertebrate, zebrafish present a considerable degree of conservation. Indeed, the zebrafish central nervous system (CNS), albeit originating from different neuro-morphogenetic processes [3], shares a similar organization with that of mammals, showing the same major brain subdivisions—forebrain, midbrain, hindbrain, and spinal cord. Similarity exists even at the cellular level, where cell types such as astrocytes [4], motor neurons [5], and cerebellar Purkinje cells [6] are closely related to their mammalian counterparts. Homologies between zebrafish and humans are not only limited to neuroanatomical and cytoarchitectonic aspects, but also extend to myelination [7] and neurochemical pathways. This organism shares the common neurotransmitter pool with mammals, and those molecules (e.g., glutamate, dopamine, serotonin, acetylcholine, glycin, GABA, etc.) exert the same biological functions as in humans [8]. In addition, multiple studies showed a conserved effect of psychoactive drugs commonly acting on human molecular targets, demonstrating the presence of shared pharmacological pathways, too [9,10]. These aspects allow many drug treatments commonly adopted in mammals to interfere with specific molecular targets to be employed in zebrafish to proficiently model phenotypic features of diverse human pathologies such as epilepsy [11] and Parkinson’s [12] and Alzheimer’s diseases [13]. Moreover, a decade ago, zebrafish genome sequencing revealed that more than 80% of genes associated with diseases in humans have an orthologous version in zebrafish [14]. This pivotal feature, combined with the ease of genetic manipulation typical of this animal model, allows for the rapid generation of mutant zebrafish lines carrying, on orthologous genes, human or de novo mutations associated with a pathological phenotype. It is thus feasible to model in zebrafish a great number of both common and rare hereditary pathologies affecting human CNS, such as epilepsies [15,16,17,18], familial Alzheimer’s disease [19], and autism spectrum disorder (ASD) [20,21], to cite just a few. Furthermore, zebrafish show homologies with humans concerning the behavioral sphere, too. Starting from a few days after fertilization and continuing through development and up to the adult stage, this organism is able to perform behaviors of increasing complexity, such as startle response and predation, to show sophisticated cognitive states, such as fear, stress and anxiety, and to perform cognitive tasks such as learning and memory recall. Interestingly, increasing evidence shows that the neural circuits controlling many of these behaviors and cognitive outputs are conserved across vertebrate classes [22,23,24], suggesting that what we investigate in zebrafish can be relevant for our comprehension of related human circuits and pathological disorders.

What we have described so far, however, are features shared by most vertebrates. Hence, what makes zebrafish an essential benchmark in today’s neuroscience? What can be performed in this organism that cannot be performed in humans or other vertebrate models such as rodents? The main answer to these questions is one: whole-brain real-time imaging at single-neuron resolution. Indeed, the possibilities glimpsed a few decades ago in this animal model sparked a massive interdisciplinary effort for the development of novel optical and biological technologies aiming at performing whole-brain functional and structural imaging in vivo. The larval zebrafish brain, which is optically transparent, is not enclosed into a bone skull and is three orders of magnitude smaller than the murine brain, represents to date the only vertebrate CNS entirely explorable in real time at submicron resolution with state-of-the-art imaging technologies. This fact, combined with the conserved features described before (from genome to behavior), represents a true game changer for neuroscience research aiming at unraveling the pathophysiological mechanisms of human disease.

In this work, we review the applications of different optical imaging techniques which have been employed to dissect the multifaceted features of human neurological disorders modeled in zebrafish. Particularly, we examine the contribution of diverse imaging approaches to decipher critical aspects of zebrafish cerebral function/structure and behavior linked to neuropathologies.

## 2. Functional Imaging

In recent years, the availability of an ever-expanding palette of genetically encoded fluorescent sensors making neuronal activity visible [25] has enhanced research efforts in developing and improving optical methods to record the said neuronal activity with high sensitivity and temporal resolution. Fluorescence microscopy, combined with state-of-the-art indicators, has emerged as an optimal tool to investigate vertebrate physiological neuronal activity and its alterations due to pathological conditions on the road towards unraveling the mechanisms ruling brain functioning. Owing to their high grade of versatility, fluorescence microscopy techniques have recently opened the possibility to explore brain functionality at multiple spatio-temporal scales.

Among the various versions of fluorescence microscopy, wide-field (epi)fluorescence microscopy (WFFM) is probably the simplest and cheapest configuration that can be attained. In this scheme, a single objective is employed to generate a uniform illumination of the sample and to collect the emitted fluorescence at the same time, while a spatially resolved detector (typically a camera) records the signal of interest (Figure 1a).

Since each frame is acquired as a single snapshot, detection speed is limited only by the camera acquisition rate and by the fluorescence photon budget. However, due to the complete lack of optical sectioning in this method, the in-focus signal is completely overwhelmed by out-of-focus contributions, with resulting images being the integration of fluorescence light coming from the whole illuminated depth of the sample. Nevertheless, this limitation in axial resolution inherent to wide-field fluorescence microscopy is typically exploited to obtain an integrated measure of the activity of entire cortex depth in mice [26,27]. Turrini and colleagues [28] first demonstrated the applicability of this microscopy technique to study the brain-wide alterations occurring in larval zebrafish CNS during pharmacologically induced seizures. Furthermore, the wide field of view obtainable with this method allowed to image the entire body of the animal, thus recording brain activity along with tail deflections, and so enabling the study of neural correlates underlying motor seizures [28] (Figure 1b).

To obtain a more thorough description of the neuronal dynamics occurring in the zebrafish brain in pathological contexts, three-dimensional (3D) imaging should be used. A prerequisite for 3D imaging is that the optical system used be capable of optical sectioning, namely the non-invasive ability to distinguish between in-focus signal and out-of-focus background. In fluorescence microscopy, two alternative strategies are typically employed to achieve optical sectioning. The first one is to illuminate the whole specimen and reject out-of-focus fluorescence (using, for example, a pinhole mask placed before the detector), as happens in confocal laser scanning microscopy (CLSM, Figure 1a). The second one is to confine the excitation only in the focus plane so as not to generate out-of-focus fluorescence contributions. This can be achieved by either exploiting non-linear interaction between light and matter to excite fluorescence only in a small volume confined at the focus of the objective, as it happens in two-photon fluorescence microscopy (TPFM, Figure 1a) [29], or illuminating the sample with a thin sheet-shaped beam: the case of light-sheet fluorescence microscopy (LSFM, Figure 1a) [30,31].

Laser scanning microscopy (LSM) comprises epifluorescence techniques which exploit the use of a focused laser rapidly pointed at different consecutive positions across a plane inside the specimen. A detector (typically a photomultiplier tube, PMT) collects the fluorescence emitted by each excited point, and a dedicated software reconstructs the image by attributing to each pixel the gray values measured by the PMT along the scanning path (Figure 1a). Due to scanning, both CLSM and TPFM present a necessary tradeoff between spatial and temporal resolution. Indeed, the more pixel-dense the image produced, the longer the time to generate it.

For this reason, apart from a few exceptions, CLSM and TPFM are typically employed for functional studies in zebrafish limited to a single medial brain plane encompassing most of the cerebral districts. Indeed, the time necessary to complete the scanning of several planes at different depths (thus performing what can be termed “volumetric imaging”) would not be compatible with functional imaging at sufficient temporal resolution for real-time neuronal activity measurements. Despite this limitation, both CLSM and TPFM are proficiently employed to study functional alterations occurring in the larval zebrafish brain in a pathological context. After the first study by Tao and colleagues [32] which employed confocal imaging to map at low spatio-temporal resolution the changes in functional connectivity occurring in the brains of zebrafish larvae expressing the ratiometric calcium indicator cameleon Y2.1 when exposed to the chemo-convulsant pentylenetetrazole (PTZ), several works followed. In fact, many other studies employed single-plane confocal imaging (yet at single-neuron resolution) on larvae expressing single-fluorescent-protein-based calcium sensors to describe the functional alterations occurring in the zebrafish brain of both pharmacological [33,34] and genetic models of epilepsy [35,36,37]. Among these works, interestingly, Liao and Kundap and colleagues [36] performed single-plane confocal calcium imaging describing the early emergence of light-induced seizures in larvae knockout for the γ2 subunit of the GABA_A_ receptor ortholog gene (Figure 1c).

Recently, a few research groups employed two-photon imaging to investigate larval zebrafish brain activity in pathological contexts. With respect to confocal microscopy, TPFM (Figure 1a), using near infrared (IR) light instead of visible light as an excitation source allows for a less invasive imaging, free from unwanted visual stimulation (IR light is by and large not perceived by the larval visual system [38]) which would be highly detrimental in delicate pathological models. Diaz Verdugo and colleagues [39] performed inspiring imaging experiments highlighting the critical role of the glia–neuron interaction in the initiation of an overt PTZ-induced seizure, with glial activity rising prior to the manifestation of an ictal event (Figure 1d). Two other groups employed single-plane TPFM in combination with local-field-potential (LFP) recordings to characterize the modifications in neuronal dynamics occurring after PTZ treatment [40,41]. Interestingly, Niemeyer and colleagues [41] employed a double transgenic line expressing GCaMP in all neuronal nuclei and dsRED in excitatory neurons so as to investigate the excitatory–inhibitory imbalance leading to seizures. As an exception to the use of two-photon imaging on a single larval medial brain plane, Hadjiabadi et al. [42] performed volumetric TPFM to reconstruct the changes in brain functional connectivity during seizures. Moreover, Andalman and colleagues [43] performing volumetric calcium imaging using TPFM identified the habenular neuronal ensembles recruited during stress encoding in the transition between active and passive coping to stressors. In addition, Haney and colleagues [44] employed volumetric TPFM functional imaging to identify the neurons of the area postrema associated with nociceptive stressors.

As anticipated, light-sheet fluorescence microscopy (LSFM) is a technique intrinsically endowed with optical sectioning. Its founding principle is the illumination of the transparent sample from the side using a thin sheet of light and the recording of the fluorescence emitted from the almost bidimensional excited plane using a second objective, its focus overlapping with the light sheet (Figure 1a). Thus, by progressively displacing the illuminated plane and coherently adapting the focus of the detection objective, the 3D reconstruction of the specimen can be obtained. Furthermore, owing to the parallelization of the detection process within each frame (each image is acquired as a single snapshot, as in WFFM), the iteration of the different illuminated planes can be achieved sufficiently fast to enable, in the most advanced setups, real-time functional imaging over the entire larval brain [45]. Despite the potential breakthrough that light-sheet imaging could bring to the investigation of the functional features characterizing different pathological zebrafish models, the full use of this technique exploiting its volumetric vocation is still in its infancy, so we expect a strong growth in the near future. LSFM was employed by Rosch and colleagues [46] to image a single medial brain plane of zebrafish larvae undergoing PTZ-induced seizures and by Winter et al. [47,48] to perform sparse volumetric sampling of the larval brain while testing the effect of several compounds on chemically induced seizures. Recently, de Vito and Turrini and colleagues [49] employed a light-sheet fluorescence microscope with double-sided non-linear excitation to investigate at high-speed and single-neuron resolution the neuronal dynamics occurring in the larval brain during PTZ-induced seizures. Besides the aforementioned advantages of employing IR light as an excitation source, the exploitation of high-speed volumetric imaging allowed the authors to describe, for the first time, a peculiar propagation pattern of ictal activity traveling in caudo-rostral direction (Figure 1e). The same group employing the same imaging system in a second work described that midbrain regions, owing to their convulsant susceptibility and early synchronous activity, could be involved in the initiation of an overt seizure [50].

**Figure 1 ijms-24-09833-f001:**
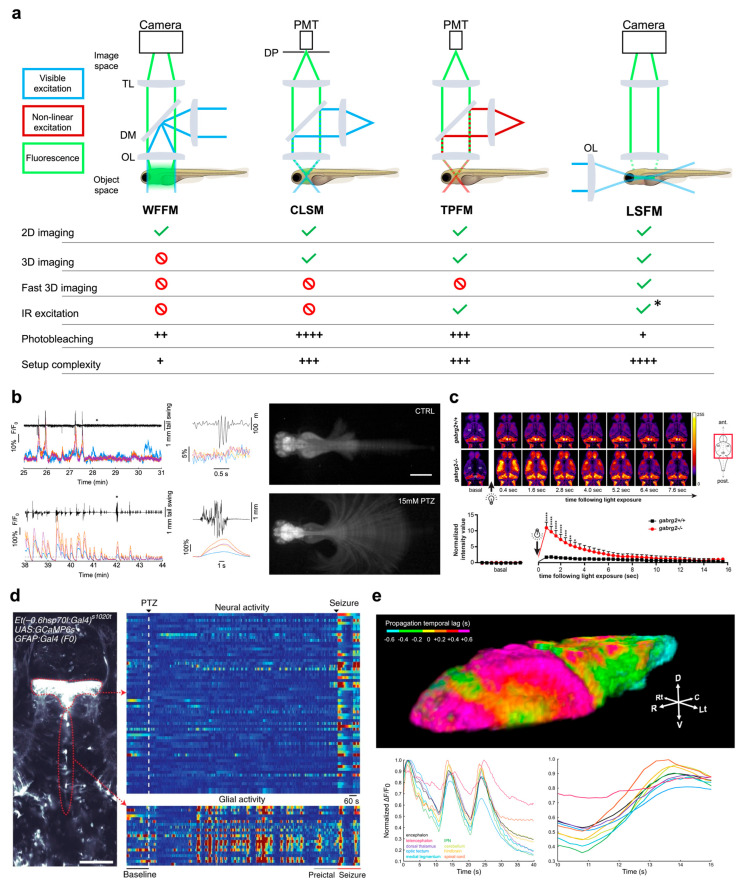
Functional imaging. (**a**) Top, schematics of the main imaging techniques adopted to perform functional imaging in zebrafish larvae: wide-field fluorescence microscopy (WFFM), confocal laser scanning microscopy (CLSM), two-photon fluorescence microscopy (TPFM), and light-sheet fluorescence microscopy (LSFM). Abbreviations: objective lens (OL), dichroic mirror (DM), tube lens (TL), photomultiplier tube (PMT), detection pinhole (DP). Bottom, table reporting the main imaging modalities achievable (green check) or not (red prohibition symbol) with each of the above techniques. The table also reports typical drawbacks (i.e., photobleaching and system complexity) quantified on a scale from one to four (black plus). Asterisk indicates that LSFM can employ IR excitation yet only in its multiphoton version. (**b**) Example of WFFM imaging performed on a pharmacological model of acute seizure. The wide field-of-view achievable with this technique enables to record calcium activity from four main brain districts (ΔF/F_0_ colored traces; telencephalon in blue, optic tectum in orange, cerebellum in yellow and hindbrain in purple) along with tail movements (black trace) in both physiological conditions (upper panels) and acute motor seizures (lower panels). Asterisks indicate the point of the traces shown, with expanded time scale, in the central panels of the figure. Images on the left represent maximum intensity projections of a motor behavior typical of physiological (upper) and pathological (lower) conditions. Scale bar: 500 μm. Figure adapted from [28], distributed under the terms of the Creative Commons Attribution License (CC-BY). (**c**) Representative single brain plane functional imaging performed using CLSM on GCaMP6f larvae expressing normal GABA_A_ receptor (upper row) and larvae knockout for the γ2 GABA_A_ receptor subunit (lower row). As visible in the temporal sequences and in the lower plot, knockout larvae (red data in the plot) show hyperexcitability to light which produces a transient seizure focused in the optic tectum. Figure taken from [36], distributed under the terms of the Creative Commons Attribution License (CC-BY). (**d**) Left, example of single-brain plane functional imaging using TPFM on larval zebrafish expressing GCaMP6s in thalamic neurons (upper red dotted outline) and periventricular glial cells (lower red dotted outline). Right, glial cells show increased activity (warmer colors) in the minutes preceding overt neuronal seizures. Scale bar: 100 μm. Figure taken from [39], distributed under the terms of the Creative Commons Attribution License (CC-BY). (**e**) Representative results obtained using 2P-LSFM whole-brain functional imaging on zebrafish larvae pan-neuronally expressing GCaMP6s. Above, 3D rendering of the temporal propagation dynamic of an ictal event during pharmacologically induced seizure. Colder and warmer colors represent regions reaching the peak of ictal activity earlier and later, respectively, with respect to the whole-brain average dynamic, as indicated by the color bar. Below, fluorescence traces highlighting the difference in temporal dynamic of the activity coming from different brain districts (color-coded as in the legend). Figure adapted with permission from [49] © The Optical Society.

## 3. Structural Imaging

Despite the fact that the functional connectivity of the brain cannot be explained only considering the anatomical substrate, it is true that brain structure and function are closely tied [51]. For this reason, the comprehension of the mechanisms underlying pathogenesis often requires an integrated approach, combining structural analyses with functional investigations. Structural alterations characterizing a pathological state of the CNS may range from the subcellular to the whole-organ level. Thus, according to the features the research aims to highlight, different techniques should be used. In addition to the imaging technique, a fundamental role in structural investigations is played by the way the sample is processed to highlight structures of interest.

The first great difference between sample processing methods is based on the need of a labeling step versus label-free methods (Figure 2a). A wide range of methods are available for the specific labeling of proteins or selected nucleic acid sequences with fluorescent or colorimetric reporters. All these techniques (e.g., immunostaining, in situ hybridization, and so on) can also be performed on the entire animal, adopting the so-called “whole-mount” protocol, with no need to mechanically section the specimen owing to the tissue transparency and the small size of larval zebrafish.

Indeed, the possibility to image, with great microscopic contrast, a fixed animal in its entirety represents an important chance for thorough structural investigations. This is why zebrafish has proven to be extremely useful in detecting cellular or anatomic distributions of proteins/mRNAs associated with the pathogenesis of diverse neurological disorders ranging from autism spectrum disorders and Alzheimer’s disease to epilepsy and myelinopathies. Pathological mechanisms of autism spectrum disorders were extensively investigated employing ex vivo staining [52,53,54,55]. Liu and colleagues [56] applied whole-mount in situ hybridization (ISH) to profile ASD-associated *shank3* transcription and its regulation by valproic acid exposure. Lüffe and colleagues [57] performed whole-mount ISH in larvae mutant for *foxp2*, a gene associated in humans to a spectrum of neurodevelopmental disorders comprehending ASD (Figure 2b). They found that *foxp2* mutation affects larval behavior through disruption of GABAergic signaling. Kozol et al. [58] employed whole-mount immunofluorescence (IF) with *pERK* staining (as a proxy for neuronal activity) in *shank3ab* mutants to demonstrate that restoring gene function rescues sensory deficits (Figure 2c). Additionally, Elsen and colleagues [59] used ISH and fluorescence in situ hybridization (FISH) to investigate the role of the *met* gene (implicated in autism in humans) during cerebellar development in zebrafish. The authors described the importance of this signaling in coordinating growth and cell type specification, functions that may underlie the correlation between altered *met* regulation and autism spectrum disorders. Furthermore, Miller et al. [60], using IF staining for the quantitation of electrical and chemical synaptic components in zebrafish larvae, found that the autism-associated gene *neurobeachin* (*nbea*) is required for both synapse formation and to maintain dendritic complexity.

Specific labeling imaging is commonly employed also in research regarding neurodegenerative pathologies such as Alzheimer’s and Parkinson’s diseases. Bhattarai et al. [61] exploited immunohistochemistry (IHC) and ISH on adult zebrafish telencephalon to demonstrate that the neuron–glia interaction mediated by both growth factors and serotonin enables regenerative neurogenesis in an Alzheimer’s disease model. Vaz and colleagues [62] employed anti-tyrosine hydroxylase IF to validate isradipine for rescuing dopaminergic cell loss in a pharmacological model of Parkinson’s disease. The same IF staining was employed by Kim et al. [63] to reveal that inhibitors of the renin–angiotensin system have a neuroprotective action in dopamine neurons.

Ex vivo specific labeling and imaging are also employed in studies regarding epilepsy. Podlasz and colleagues [64] employed IF on adult zebrafish brain to localize the production of galanin, a neuropeptide whose overexpression is correlated with potent anticonvulsant effect in the zebrafish PTZ seizure model. Interestingly, galanin was investigated through IHC and ISH by Corradi et al. [65] to identify its role in stress regulation. They determined that, in the hypothalamus of zebrafish larvae, galanin has a self-inhibitory action on galanin-producing neurons, playing an important role in the prevention of potentially harmful overactivation of stress-regulating circuits.

Whole-mount IF and FISH are also used to study the pathological mechanisms underlying myelinopathies. With those techniques, Zada and colleagues [66] monitored myelin-related processes and structural synaptic plasticity in a zebrafish model for psychomotor retardation.

Labeling of the structure of interest can also be achieved through genetic encoding (Figure 2a). Owing to the ease of genetic handling, countless transgenic zebrafish reporter lines have been generated [67]. Those strains typically express, under specific promoters, a fluorescent protein or a fusion protein composed of the protein target of the study (transcription factor, soluble protein, membrane channel, etc.) and a fluorescent protein tag. The use of reporter lines enabled the longitudinal imaging of an astonishing number of signaling pathways. Reporter lines have been so far applied to the investigation of pathophysiological mechanisms in many disease models. Paquet et al. [68] generated a reporter line carrying the human *tau* protein gene fused to the DsRed gene as a model of tauopathy recapitulating several clinical features of neurodegenerative diseases such as Alzeimer’s disease and chronic traumatic encephalopathy [69]. This line was employed to study the mechanisms leading to pathological tau protein aggregation [70,71], microglial activity against tauopathic neurons [72] and the role of brain-derived neurotrophic factor signaling in tauopathy progression [73].

Imaging on transgenic reporter lines proved useful also in the investigation of ASD models. Jamadagni et al. [74] employed a strain expressing GFP in GABAergic interneurons on a background mutant for the *chd7* gene responsible for CHARGE syndrome, an ASD-related disorder. They showed that *chd7* functions are required for the correct development of GABAergic neurons. Interestingly, a reduction in GABAergic neurons was reported also by Hoffman and colleagues [20] on a different genetic background (autism risk gene *cntnap2*) using transgenic expression of GFP under the GABAergic promoter *dlx5a/6a* (Figure 2d). The pan-neuronal promoter HuC was exploited for generating transgenic lines to investigate the way in which the autism-related genes *met* [59] and *shank3b* [21] affect brain development.

Reporter lines have been applied also to study myelination. Jung and colleagues [75] generated the transgenic line *Tg(mbp:gfp)*, specifically tagging oligodendrocytes and Schwann cells with GFP (under the control of myelin basic protein promoter), thus allowing the visualization of myelin sheaths from embryos to adults. This line was employed to identify compounds promoting myelination [76], served as a background for chemogenetic demyelination zebrafish models [77], allowed to identify an enzyme involved in the synthesis of sphingolipids crucial for myelination [78], and was used to assess efficacy of targeted gene therapy in a model of hypomyelination [79]. Xiao et al. [80] selectively expressed GFP in Schwann cells to study axon–glia interactions during the repair process in a zebrafish model of peripheral nerve injury.

Reporter lines have proved particularly useful in the investigation of the pathological mechanisms leading to Parkinson’s disease [81,82] as well. Xi and colleagues [81] produced the transgenic line *Tg(dat:EGFP)* expressing EGFP gene under the dopamine transporter promoter. The in vivo tagging of dopaminergic neurons allowed, for example, the study of L-DOPA selective toxicity [83]. Moreover, Godoy and colleagues [84] used this line as a background to develop a chemogenetic ablation model of dopaminergic neurons. Weston et al. [85] expressed human α-synuclein in zebrafish larvae as a fusion protein with GFP to study its presynaptic aggregation. Interestingly, Lopez and colleagues [86] employed zebrafish larvae expressing α-synuclein fused to the photoconvertible fluorescent protein Dendra2 to investigate the α-synuclein clearance through longitudinal imaging.

Specific applications for the study of zebrafish models of human diseases at very high resolution and magnification have recurred to transmission electron microscopy (TEM, Figure 2a). TEM employs an accelerated beam of electrons which is focused through magnetic lenses onto an ultrathin section of the specimen, normally endowed with contrast for biologically relevant features (e.g., membranes or other targeted structures) by staining with heavy metals (typically Pb and/or U). Electrons passing through the samples (i.e., transmitted electrons) are collected by a detector to form an image where stained electron-dense structures appear darker while unstained electron-lucent ones appear lighter [87]. Early and colleagues [76] employed TEM imaging to evaluate the myelination grade of the axonal projection in the larval spinal cord (Figure 2e). Similarly, Turcotte et al. [88] used TEM to highlight the increasing coherence of myelin sheaths during zebrafish development. Aspatwar et al. [89] using TEM imaging investigated the apoptotic effects of mutation of the carbonic anhydrase-related protein VIII gene, associated in humans with mental retardation and ataxia.

As we previously mentioned, there are also structural imaging techniques which do not require any labeling of the structures of interest. Interestingly, these optical methods offer the ability to image living zebrafish embryos, often at high spatial resolution and with reduced or absent photodamage. Among those techniques, optical coherence tomography (OCT, Figure 2a), originally developed for ophthalmological applications, has proven to be a powerful tool in various biomedical research fields [90]. OCT optimally works on translucent and thin specimens, and this is the reason why it is successfully applied to zebrafish imaging. OCT represents a non-invasive method to perform in vivo structural investigations in both the larval and adult zebrafish CNS [91,92] and was applied to diverse pathological models ranging from brain tumors [93] (Figure 2f) to β-amyloid induced brain atrophy [94] and chemically induced notochord developmental defects [95].

**Figure 2 ijms-24-09833-f002:**
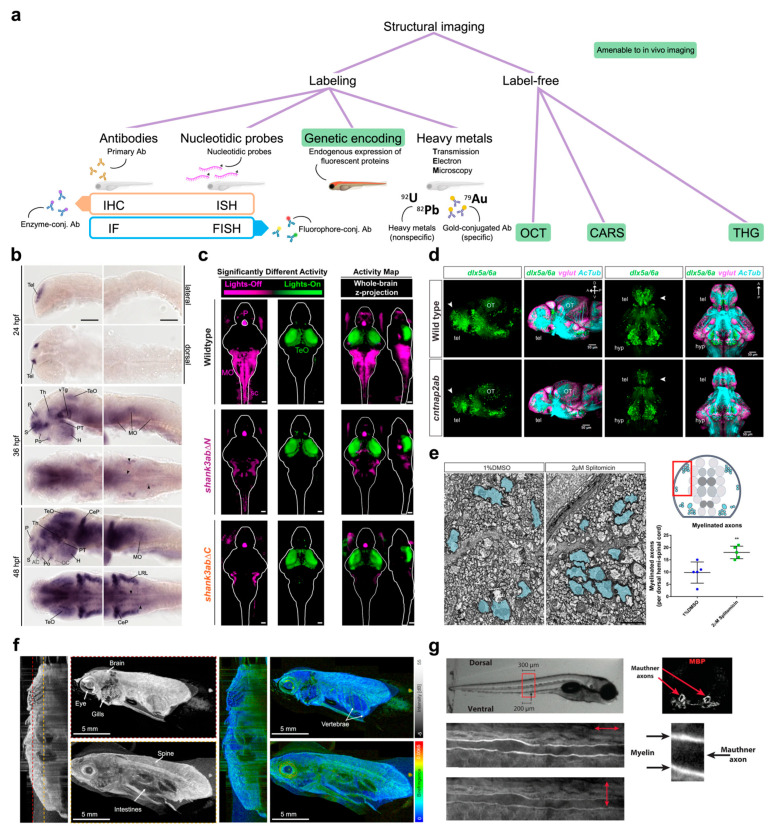
Structural imaging. (**a**) Tree diagram reporting a classification of structural imaging methods. Abbreviations: antibodies (Ab), immunohistochemistry (IHC), immunofluorescence (IF), in situ hybridization (ISH), fluorescence in situ hybridization (FISH), optical coherence tomography (OCT), coherent anti-Stokes Raman scattering (CARS), third harmonic generation (THG). (**b**) Representative whole-mount ISH showing the spatial distribution of *foxp2*-positive neuronal populations into the developing zebrafish brain. Scale bars: 100 μm. Figure taken from [57], distributed under the terms of the Creative Commons Attribution License (CC-BY). (**c**) Activity maps (*z*-projections) obtained from whole mount immunofluorescence staining of *pERK*, used as a proxy of neuronal activity. Magenta indicates neurons with increased activity in the lights-off transition while green indicates those showing increased activity during lights-on transition. Compared to wild type, *shank3ab* mutants show normal activation of the pineal gland (P) but fail to show activation of the medulla oblongata (MO) and spinal cord (sc) during the lights-off transition. This indicates that *shank3ab* mutant models sense light normally but fail to activate downstream brain regions underlying sensorimotor integration. Scale bars: 50 μm. Figure taken from [58], distributed under the terms of the Creative Commons Attribution License (CC-BY). (**d**) Reporter line showing GABAergic (green) and glutamatergic (magenta) neurons and axonal projections (cyan) on both wild type and *cntnap2ab* mutant. The autism risk gene produces a significant reduction in GABAergic neurons in the telencephalon (white arrowheads). Figure reprinted from [20], with permission from Elsevier (please see *Acknowledgements* for details). (**e**) TEM micrographs of larval zebrafish spinal cord portion showing the increase in myelinated axons upon treatment with splitomicin. Red box in the cartoon on the right indicates the approximate spinal region where the micrographs were taken. Scale bar: 1 μm. Figure taken from [76], distributed under the terms of the Creative Commons Attribution License (CC-BY). (**f**) Representative polarization sensitive OCT acquisition of an adult zebrafish showing intensity (grays) and birefringence (colors) signals. Birefringence signal provides a measure of tissue organization (warmer colors, higher level of organization). For both signals, a sagittal cross section and two en face images at the position indicated by the dotted lines are presented. Figure adapted from [92], distributed under the terms of the Creative Commons Attribution License (CC-BY). (**g**) Polarization-resolved CARS imaging of a zebrafish larva allows the visualization of Mauthner axons. Different light polarizations (as indicated by double headed arrows) differently modulate CARS signals of myelin sheaths. Figure taken from [88], distributed under the terms of the Creative Commons Attribution License (CC-BY).

Another label-free optical technique enabling in vivo structural imaging is coherent anti-Stokes Raman scattering (CARS, Figure 2a) [96]. CARS signal is generated from the stimulated vibrational motion of molecular bonds in the sample. In its most common two-color variant, two infrared lasers, a “pump” beam and a Stokes beam, interact with the sample to selectively and coherently excite specific molecular bonds. A third photon (usually from the pump beam) interacts with the excited bonds to induce their relaxation back to the ground state. This step is accompanied by the emission of a fourth photon (anti-Stokes emission) which represents the detected signal. CARS microscopy enables imaging based on chemical contrast, guaranteeing long-term observation without photodamage or photobleaching effects. Turcotte and colleagues [88] performed polarization-resolved CARS imaging in zebrafish larvae to investigate promyelinating treatments on reticulospinal neurons axons (Figure 2g). This CARS variant enables the all-optical label-free detection of the degree of order of the myelin sheath molecular architecture [97], a quantity that is linked to myelin health [98,99].

Among label-free imaging techniques applied on the transparent larval zebrafish brain, it is also worth mentioning the third harmonic generation (THG, Figure 2a). THG microscopy, using infrared lasers, exploits the light–matter nonlinear interaction properties which induce the generation of a new component of the electromagnetic field at the focal point of a laser beam passing through local sharp transitions in third-order nonlinear susceptibility or refractive index. The new wave is produced in the forward direction and is characterized by a wavelength equal to a third of the wavelength of the illumination beam [100]. The THG signal is typically produced in biological samples by water–lipid and water–protein interfaces. THG imaging was employed in zebrafish larvae, for example, to study neural development [101,102] and myelination [103].

The imaging approaches discussed so far in this section represent the main methodologies applied to date to investigate the structural features of zebrafish CNS in the context of pathological models. Despite not yet been reported as being applied to investigate specific zebrafish cytoarchitectonic aspects in pathological contexts, there are a few other imaging techniques which are worth mentioning owing to their potential. Super resolution microscopy (SRM) [104] includes several fluorescence imaging techniques linked by the ability to surpass the diffraction limit of optical resolution. SRM has progressively pulled down the spatial resolution barrier towards nanometer scales, enabling imaging of cellular structures with a level of detail that was previously exclusive to electron microscopy. Yet, compared to the latter, SRM methods retain the advantages of fluorescence microscopy both in terms of target specificity and sample preservation, thus also enabling in vivo imaging. Among SRM techniques, structured illumination microscopy (SIM) [104] and its derivatives have been employed to study synaptic structures of spinal projection neurons [105], microtubule organization in midbrain and hindbrain neurons [106] and the posterior lateral line primordium [107] in living transgenic zebrafish larvae and embryos. A different type of SRM, stimulated emission depletion (STED) [108] microscopy, was instead employed to investigate the spatial localization of different ribbon synapse proteins in zebrafish larvae after IF staining [109]. Moreover, among the label-free imaging methods enabling investigations about structural features of neural tissue, it is also worth mentioning micro computed tomography (micro-CT) [110] and matrix-assisted laser desorption/ionization mass spectroscopy imaging (MALDI-MSI) [111]. Micro-CT is the analogue of clinical CT, yet on a smaller spatial scale and with greatly increased spatial resolution. It employs X-rays and sample rotation to obtain submicrometric 3D reconstructions of samples that are up to 20 cm wide. Micro-CT was proficiently employed to reconstruct volumetric whole-body structures of juvenile zebrafish and to count neuronal nuclei in the entire brain of zebrafish larvae [112]. On the other hand, MALDI mass spectroscopy imaging employs a laser to perform a pixel-wise scan of thin sample slices. In this way, it is able to produce a compound-specific mass spectrum map, where pixel intensities reflect the local abundance of a detected analyte. This technique, capable of analyzing the molecular composition and spatial distribution in biological samples, has been used to assess potential brain accumulation of anticancer agent compounds [113] and synthetic cannabinoid [114] in zebrafish larvae and to investigate clozapine brain metabolism in adult zebrafish [115].

## 4. Behavioral Imaging

The ability of zebrafish larvae to perform several behaviors appears early during ontogenesis (as early as 17 h after fertilization, zebrafish embryos produce typical coiling movements inside the chorion [116]). Therefore, behavior has been widely employed to characterize zebrafish phenotype in diverse pathological models, with a multitude of behavioral imaging methods designed according to the specific aims. Indeed, due to important changes in size and behavioral repertoires from larval to adult stage, the setups are typically designed for a specific developmental window and sometimes even for a specific behavior. These systems typically comprise a behavioral arena, an illumination system (visible and/or infrared) and one or more imaging cameras. Arenas can have many different shapes specifically designed to evaluate peculiar aspects of behavior.

Starting from the larval stage, probably the most widely diffused behavioral imaging method is represented by high-throughput imaging systems (Figure 3a). Whether custom [28,117] or commercial, these setups generally have a wide-angle lens capable of a large enough field-of-view to image the surface of a conventional multi-well plate (>80 cm^2^). High-throughput systems are typically employed to assess larval locomotor activity on as many as 96 larvae in parallel. Moreover, owing to their parallelization and segregation of individual larvae in separate wells, these systems are very effective for preliminary screening on large libraries of drugs. This approach allows brute force testing of a huge number of compounds without any prior knowledge on their effects, followed by more in-depth investigations on the most promising candidates [118]. It should be noted that this approach is radically different, at least in the initial selection step, from traditional drug design approaches and it has opened new and complementary pathways in drug discovery. The behavioral parameters obtained are typically aggregated measurements (such as average speed, total distance traveled, time spent in movement, etc.) along the entire recording duration or temporal sub-windows, useful to identify quantitative differences in larval locomotor activity. Out-of-the-box solutions (such as DanioVision by Noldus and ZebraBox by ViewPoint) come with proprietary software which automatize several analysis steps, thus significantly reducing post-processing time. On the other hand, custom solutions are typically much less expensive and much more versatile yet requiring basic technological knowledge to be set up. High-throughput systems are applied to investigate alterations in larval locomotor activity in many different models recapitulating salient features of human brain diseases such as autism spectrum disorders [20,21,57,74], myelinopathies [78], Parkinson’s disease [63,82,119], psychiatric disorders [89] and anxiety-related disorders [120].

In particular, Hoffman and colleagues [20] identified, among 550 psychoactive screened molecules, the phytoestrogen biochanin A to reverse the nocturnal hyperactivity phenotype in an ASD-associated mutant zebrafish line. Notably, Vaz and colleagues [62] screened 1600 bioactive drugs on a 6-hydroxydopamine zebrafish model of Parkinson’s disease and determined isradipine to proficiently rescue the bradykinetic and dyskinetic-like behaviors. Richendrfer et al. [120] employed a custom high-throughput assay coupled to a visually induced stressor (bouncing balls presented on a display beneath the arena) to characterize anxiety-related behaviors in zebrafish larvae. Moreover, high-throughput assays are widely employed in epilepsy research, both in behavioral characterizations and in drug screening assays. Wasilewska and colleagues [121] reported that a zebrafish line carrying a human mutation (*stxbp1* gene) associated with epilepsy shows spontaneous seizures and increased sensitivity to PTZ effects. Suo et al. [122] found that a mutation in the *stim2b* gene (encoding a protein involved in the regulation of store-operated Ca^2+^ entry) induces increased locomotor activity, thigmotaxis, PTZ and glutamate susceptibility as well as disruption of physiological phototaxis in zebrafish larvae. Dinday and Baraban [123], after screening more than a thousand compounds, determined that dimethadione is able to suppress the behavioral seizure component in a zebrafish genetic model of Dravet syndrome. Interestingly, Turrini and colleagues [28] devised an optical system combining high-throughput behavioral recording (yet with a custom honeycomb arrangement of the wells) with the integrated recording of brain calcium transients in parallel on 60 larvae pan-neuronally expressing GCaMP6s. With this system tested on the PTZ model of seizures, they demonstrated an increased selectivity in identifying the effects of different compounds with respect to behavioral recording alone (Figure 3b).

A finer and more thorough description of larval swimming often requires recording individual animals with the possibility to test the effects of non-chemical stimulations on behavior. In fact, zebrafish larvae are sensitive to a variety of stimulus modalities, including touch, auditory and vestibular inputs, heat, and vision.

With respect to high-throughput setups, these systems often have a single larger behavioral arena and additional components such as a display for visual stimulation [50], ambient lights of multiple colors [43] or devices for tactile stimulations [124] (Figure 3a). Sometimes, these systems are equipped with cameras that have enhanced performances, enabling high-speed tracking necessary to characterize larval swimming kinematic. Despite the small size, zebrafish larvae can exhibit a peak acceleration of 20,000 mm/s^2^ and a peak velocity of 200 mm/s [125], thus representing a demanding task in terms of tracking capabilities. Gauthier and colleagues [124] reported that zebrafish larvae mutant for the autism- and schizophrenia-associated gene *shank3* show defective or reduced touch-induced escape response. Similar results were presented by Miller and colleagues [60] on a zebrafish line mutant for the autism-associated gene *nbea*. Andalman and colleagues [43] exploited a larval tracking system to show that animals exposed to stressors (electric current) make a transition from active to passive behavior (Figure 3c). A similar result was obtained by Haney et al. [44] who investigated the effects of a nociceptive stressor on zebrafish larvae light–dark preference, reporting that prolonged stressor exposure suppresses larval typical exploratory behavior. Turrini and colleagues [50] employed high-speed tracking (at 300 fps) to describe the critical features of larval swimming kinematic at increasing seizure stage. The system they used performs imaging in the infrared wavelength range, thus also allowing visual stimulations employing a display underneath the arena without unwanted visual interference.

As larvae grow, their cognitive abilities improve, thus making the larvae amenable to be employed in a variety of tests aiming at evaluating more complex behavioral manifestations such as learning and memory, social preference, and anxiety, to cite just a few. In this type of experiments usually performed on later-stage larvae (21–30 days post-fertilization—dpf), juveniles (30–90 dpf) and adults (>90 dpf), custom behavioral arenas are typically employed. The shape of the arenas is typically a maze specifically designed to evaluate a certain behavior (Figure 3a). While T-maze [126] and Y-maze [127] are usually employed to assess learning and memory [128] as well as place preference [129], plus-maze [130] is predominantly used to evaluate anxiety-related behaviors [131]. Interestingly, some of these mazes also allow the study of social preference, which is pivotal in those neuropathologies linked to alterations in social interactions. Amongst all shapes, three-chambers maze [132] is the one that better enables the investigation of social preference in zebrafish [133]. Owing to its shape, three-chambers maze allows an accurate quantification of the time an animal spends near to or away from its fellows. In addition to these maze types, there exist peculiar multi-chamber shapes, the so-called alternative-maze [134] shapes typically designed to evaluate learning and memory skills (Figure 3a).

Zimmermann and colleagues [135] employed a three-chamber maze to show that embryological treatment with valproic acid induces deficit in social interaction, anxiety, and hyperactivity both in juvenile (70 dpf) and adult zebrafish (120 dpf). Similarly, Dwivedi et al. [136] tested social preferences on 21 dpf larvae early-treated with valproic acid employing an alternative maze with an “s” shape. Liu et al. [21] and Zheng et al. [55] reported that zebrafish mutant for genes *shank3* and *katnal2*, respectively, when tested for social preference in a three-chamber maze showed autism-like behaviors (Figure 3d). Fulcher and colleagues [137] employed a three-chamber maze on adult zebrafish subjected to unpredictable chronic mild stress following developmental isolation (model of major depression) to evaluate social preference. Kumari and colleagues [138] employed T-maze and three-chamber maze on a pharmacological model of chemical kindling in adult zebrafish, highlighting that the kindling-like state alters spatial cognition but not social novelty recognition. Jarosova and colleagues [139] employed an adult zebrafish pharmacological model of Alzheimer’s disease (okadaic acid) showing impaired learning and decreased motivation in reaching the goal chamber in an alternative maze. This peculiar type of maze was also employed by the same group to describe the locomotor and cognitive deficiencies in a rotenone Parkinson’s disease model in adult zebrafish [140] (Figure 3e). Cleal and colleagues [141] tested 24-month-old zebrafish performance in a Y-maze, demonstrating that the aging-related cognitive decline is related to dopaminergic activity.

Starting from the juvenile stage (>30 dpf), zebrafish produce a significant modification of their swimming behavior. Indeed, if during the larval stage zebrafish tend to occupy the layers of water closer to the bottom (excluding foraging periods), from the juvenile stage, they permanently occupy the entire water column. From this stage, apart from experiments using mazes with shallow waters, complete behavioral imaging cannot rely on a single camera placed on top of the arena. The use of a second camera, placed orthogonally to the first, allows the discrimination of fish movements in the water column. Thus, 3D behavioral imaging systems (Figure 3a) not only enable the fine description of zebrafish swimming behavior in their environment (such as their home tank, Figure 3f), but also allow the characterization of complex adult social behaviors (and their alterations) such as shoaling, aggression or mating [142]. Furthermore, this experimental scheme also enables specific behavioral assays, such as the novel tank diving test which uses the vertical distribution in a novel environment as a measurement of anxiety-like behaviors in adult zebrafish [143]. Maaswinkel and colleagues [144] employed 3D behavioral imaging to track the shoaling behavior of adult zebrafish. They highlighted that the treatment of a single fish with an NMDA receptor antagonist, previously shown to mimic aspects of autism and schizophrenia (MK-801), drastically reduces the shoal cohesion (Figure 3g). The same group, employing the same experimental apparatus, reported an increase in anxiety, locomotor activity and stereotypy in an adult zebrafish knockout for the *fmr1* gene, model of fragile X syndrome [145]. Zheng et al. [55] performed behavioral imaging on adult zebrafish knockout for the ASD-associated gene *katnal2*, describing increased thigmotaxis and geotaxis, as well as peculiar swimming behaviors such as big circling, small circling, walling, and cornering. Similar results were obtained by Liu et al. [21] on a *shank3b* knockout line.

**Figure 3 ijms-24-09833-f003:**
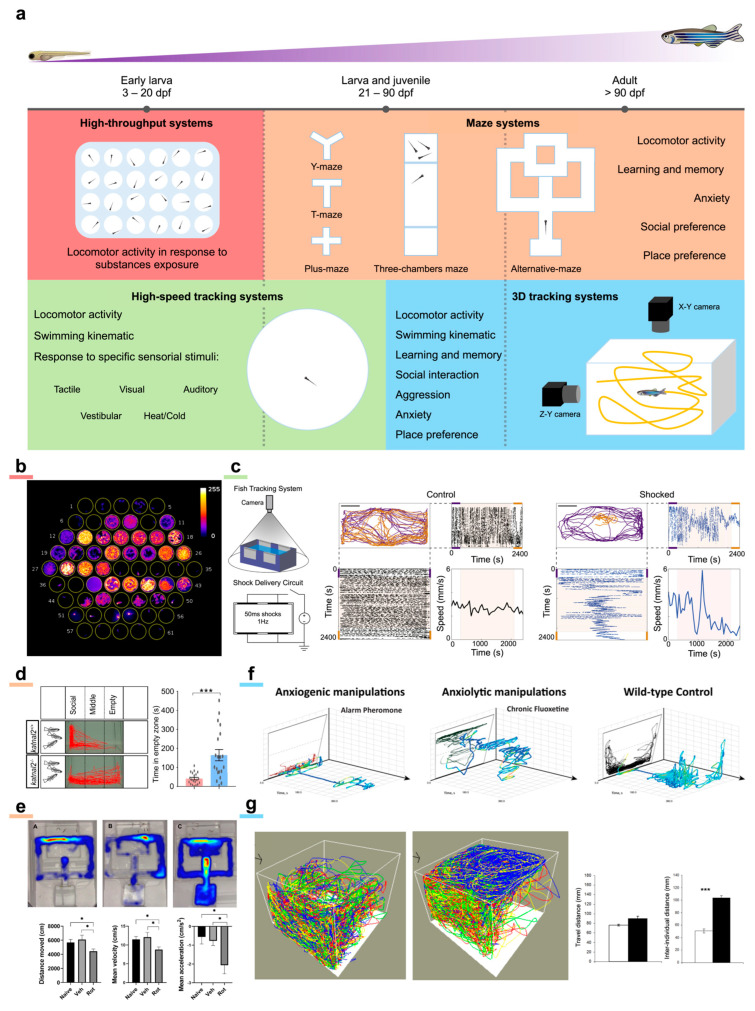
Behavioral imaging. (**a**) Schematic representation of the manifold behavioral imaging methods employed on zebrafish depending on the developmental stage and on the behavioral features to be studied. (**b**) High-throughput behavioral assay combining measurement of locomotor activity and calcium activity of the brain of 60 GCaMP6s larvae exposed to different pharmacological treatments in combination with PTZ. Image is a maximum intensity projection of a 13 min recording. Warmer colors indicate higher brain activity. Figure taken from [28], distributed under the terms of the Creative Commons Attribution License (CC-BY). (**c**) Behavioral imaging system specifically designed to study active/passive coping behavior in response to a stressor (electrical shock). Larval trajectories before (purple) and after (orange) shock are presented. The stressor produces an active-to-passive coping. Figure reprinted from [43], with permission from Elsevier (please see *Acknowledgements* for details). (**d**) Example of a three-chamber maze employed to evaluate social preference in a zebrafish line knockout for the gene *katnal2*. Mutant fish show impaired social preference with respect to wild type animals. Figure taken from [55], distributed under the terms of the Creative Commons Attribution License (CC-BY). (**e**) Example of an alternative maze employed on adult zebrafish treated with rotenone as a model of Parkinson’s disease. Heatmap overlays (the warmer the colors, the longer the time spent in that position of the maze) show that treated animals (third map from left) spent more time in the close arm than in the goal chamber with respect to untreated and vehicle animals (first and second map from left, respectively), indicating a decrease in cognitive abilities. Plots show an overall reduction in locomotor activity in treated animals with respect to both untreated and vehicle fish. Figure reprinted with permission from [140] © American Chemical Society. (**f**) Representative 3D tracking of adult zebrafish exposed to either anxiogenic (first plot from left) or anxiolytic (second plot from left) treatments. Fish treated with anxiogenic compound (alarm pheromone) display typical geotaxis behavior. Trajectories are color-mapped according to fish swim velocity (warmer colors indicate higher velocities). Figure taken from [142], distributed under the terms of the Creative Commons Attribution License (CC-BY). (**g**) Example of 3D trajectories of shoaling adult zebrafish. The first plot from left shows in different colors the swimming trajectories of a homogeneous quadruplet of control fish. The second plot shows trajectories of a heterogeneous quadruplet composed by three control fish and one zebrafish (blue line) treated with a compound mimicking behavioral features of autism/schizophrenia. Heterogeneous quadruplet does not show increased locomotor activity with respect to a homogeneous one. However, treated fish swim far apart from control ones and typically also produce a reduced cohesion of the shoal (increased interindividual distance in the heterogeneous quadruplet). In the bar plot, white refers to a homogeneous shoal while black to a heterogeneous one. Figure adapted from [144], distributed under the terms of the Creative Commons Attribution License (CC-BY).

## 5. Discussion

Functional, structural, and behavioral optical imaging comprehends a series of technologies which have been revolutionizing the approach to biomedical research and in particular to the field of neuroscience both for basic research and for the study of models of human diseases. Part of the credit for this success is undoubtedly due to the ever-increasing adoption of zebrafish as an animal model. This small vertebrate, owing to its unique features, has fostered research efforts in developing novel technologies, both optical and biological, able to exploit the full research potential of this organism.

Among the functional imaging applications we reviewed in this paper, light-sheet fluorescence microscopy plays a crucial role in zebrafish research. Indeed, this microscopy technique whose architecture was conceived at the very beginning of the twentieth century has been at the heart of a true renaissance which started in the last decade of the twenty-first century thanks to the combination with the larval zebrafish model. In fact, we can say that LSFM found its perfect match in this tiny vertebrate, to the point that after a few decades this technique has benefited from many as well as dramatic improvements [146,147,148,149,150,151,152] which contributed to pushing further its spatio-temporal resolution abilities until reaching unparalleled mapping capabilities of the entire larval brain in real time [45,49,153]. So far, LSFM, and functional imaging techniques in general, in pathological contexts have been mainly applied to the investigation of epilepsy models. If on the one hand this aspect depends on paroxysmal alterations of neuronal activity being the specific signature of epilepsy, on the other hand, a lot of research on other pathologies such as autism spectrum disorders could strongly benefit from fast whole-brain functional investigations. Moreover, in recent years, cellular resolution whole-brain atlases of larval zebrafish neuroanatomy have been set up [154,155]. These informatic tools are providing precious help both in the neuroanatomical identification of functional areas and in comparing data obtained across diverse laboratories with different imaging techniques.

In this context, optical and biological methods for the detailed study of structural aspects of the larval brain did not lag behind. The production of reporter lines expressing fluorescent proteins in a cell/time-specific manner provided the unprecedented possibility to see biological processes and signaling pathways happening in the entire CNS of a living vertebrate [67]. While the use of these transgenic lines opened the way to longitudinally investigate pathological alterations by common confocal imaging, ex vivo investigations employing diverse labeling methods have undergone important advancements as well. In recent years, there has been a great interest around the specific labeling of protein/nucleic targets in zebrafish. Proof of this is the fact that several companies have flourished offering antibodies specifically developed to bind zebrafish isoform targets. Moreover, structural investigations exploiting fluorescent tags (e.g., fluorescent proteins, fluorophore-conjugated antibodies/nucleotidic probes, etc.) result extremely versatile since they can be performed using a wide range of fluorescence microscopy setups (such as WFFM, CLSM, TPFM and LSFM) also employed in functional imaging applications. In addition to this, great efforts have been recently made to produce synaptic resolution reconstructions of the structural connectivity of the larval zebrafish brain using electron microscopy [156]. Furthermore, the combination of brain-wide functional imaging with nanometric structural descriptions of the entire organ [157] represents a unicum for vertebrates and could be a game-changing approach in the investigation of the pathological structural alterations characterizing brain disorders.

In the framework of imaging approaches, both larval and adult zebrafish lend themselves to the study of pathological alterations of behavior. The extensive behavioral repertoire of zebrafish enables the investigation of phenotypic aberrations typical of pathological contexts ranging from epilepsy and autism spectrum disorders to Parkinson’s and Alzheimer’s diseases. While the high-throughput capabilities enabled by larval small size have been proficiently applied to discover novel promising therapies, high-speed tracking and the kinematic analysis of swimming [158] so far have not been particularly employed in neurological disease models while representing a promising approach.

## 6. Conclusions

In this review, we discuss several imaging approaches which are increasingly used in zebrafish research on neurological diseases. Optical techniques applied to this animal model offer a unique chance for translational neuroscience research at different scales ranging from the whole organism (behavioral) to brain activity (functional) and even its subcellular alterations (structural). Despite the ever-increasing adoption of optical methods to investigate pathological mechanisms in zebrafish, the impression is that the enormous technological improvement we have witnessed has yet to make a complete transition from basic to applied research. Indeed, we deem that in the next decade, the widespread use of the most advanced methods we discussed, together with the use of groundbreaking technologies such as genetically encoded fluorescent sensors for neurotransmitters imaging [159,160,161], optogenetic intervention [162,163,164] and whole-brain functional imaging in freely-swimming larvae [125,165] will foster a deeper understanding of pathophysiological mechanisms underlying translational zebrafish neuropathological models.

## Data Availability

Not applicable.

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
