# Peer review of "Imaging Approaches to Investigate Pathophysiological Mechanisms of Brain Disease in Zebrafish"

_ijms, 2023, doi:10.3390/ijms24129833_

Round 1

Reviewer 1 Report

This review is interesting because it indicates the different approaches to resolving images of zebrafish regarding the study of neurological disorders. It is necessary to introduce some clarity before publishing this paper. I have some suggestions:

- I invite the authors to define better the title. Why have you used the word "models"? Is it relative to zebrafish as an experimental model or to the different applied techniques?

- The authors present this paper as a review but then in the abstract, they explain that “In this review, we present an overview of imaging approaches…..” Have review and overview similar significate or with the term overview the authors want to reflect the limit of a summary?

 - English language is correct in complex but in some phrases, I invite you to revise the text. For example:

a) in Abstract …“Owing to its peculiar features and high grade of genomic homology with humans, it is increasingly employed to model diverse human neurological disorders, both through genetic and pharmacological intervention.”…  Human is repeated but it is not necessary.

b) “Indeed” is repeated too many times.

c) There are several phrases much long to define better.

d) Is possible to explain better the concept of volumetric imaging?

e) For the TEM technique, the authors give few clarifications, unlike other techniques. I suggest giving some other important indications.

English language is correct in complex but in some phrases, I invite you to revise the text. For example:

a) in Abstract …“Owing to its peculiar features and high grade of genomic homology with humans, it is increasingly employed to model diverse human neurological disorders, both through genetic and pharmacological intervention.”…  Human is repeated but it is not necessary.

b) “Indeed” is repeated too many times.

c) There are several phrases much long to define better.

Reviewer 2 Report

Turrini et al. presented a fascinating review on zebrafish imaging related to brain diseases. As a basic medicine researcher, I think it would be worth publishing such an article in International Journal of Molecular Science, because it is well prepared and the findings are exciting. The introduction and literature search method to summarize the past efforts is thorough. However, several inconsistencies throughout the article have been noticed. I suggest the authors to find below a couple of comments and concerns that they should address-

Major concerns:

1.        Figure 1 is reproduced without appropriate permission. As authors are still waiting for formal permission, it is better to replace or omit this figure if it is not possible to get the permission to avoid copyright issues.

2. In figure 3g, description of white and black bars are unavailable in figure/legend. Either include it in the figure or at least in the figure legend.

3. I recommend the authors to use a reference management software package like EndNote or Mendeley if they did not use it for this manuscript because reference styles are inconsistent.

 Minor concerns:

1.        In the abstract section line, it would be better to avoid the word “peculiar”. It may sounds like a negative impression to some readers. I recommend the author replace with other word, for example “distinctive”.

2.        If possible I encourage the authors to expand the area of imaging modalities by including micro-CT (For example, DOI: 10.7554/eLife.44898), mass spextrometry imaging (for example , DOI: 10.1038/s41598-022-09659-y ) etc.

English laguage quality is fine except few corrections.

Reviewer 3 Report

Dear editor and authors,

the manuscript entitled „Imaging approaches to investigate pathophysiological mechanisms of brain disease in zebrafish models” describes the possible approaches of imaging techniques nowadays used in the widely distributed model organism zebrafish. After brief introduction where the usefulness of the zebrafish model organism in comparison to other models is highlighted (size, whole brain imaging, transparent and so on) and a brief mention of available models for specific brain disorders, the authors group the described imaging techniques into three distinct categories. They start with so called “functional imaging” where they group the classical microscopy techniques together which are used in fluorescence microscopy (wide-field, confocal laser-scanning, two-photon, light sheet) and give examples of zebrafish studies using the discussed technique. The second group is named “structural imaging” where the labeling techniques used is in the focus. They distinguish between label-free techniques for imaging (like OCT, CARS and THG, which was absolutely new to me) and specific labeling techniques (Ab, insitu, reporter lines, Calcium-indicators, and so on) and introduce other microscopy techniques as well (like TEM). The third part sums up the possibilities for behavioral imaging techniques. A wide variety of age-specific assays and imaging approaches together with a multitude of examples from the literature concludes this section. A brief discussion section and conclusion close this review.

This manuscript is well written and cites a multitude of relevant studies in the field. Although, I would have chosen some other papers on some occasions, the choice the authors made is absolutely reasonable. There were several papers I was not aware of and which I will have to read soon. Thanks for that. The quality of this review is very high, and it deserves publishing. Well done.

The only real criticism I have is the structure of this review in one particular aspect. The distinction of structural and functional imaging is not really clear to me. For sure, there is a substantial overlap in both chapters. In the functional imaging section (chapter 2) the authors mainly describe the microscopic technique used and cite studies in which these techniques have been used. The next chapter (3 called structural imaging) focuses on labeling techniques which are mainly using the same microscopy techniques described in chapter 2 (e.g. for visualizing an Ab staining you could use CLSM, or wide-field or LSM). In principle this is fine, but then I would call the chapters differently. Or maybe a different grouping of the studies and topics would make more sense. Focus on the microscopy or other imaging technique (wide-field, confocal, and so on, but include in addition super-resolution techniques like STORM and EM techniques and OCT and CARS could also make sense here) could be done in chapter 2 (Maybe call imaging techniques?). Use the specific labeling technique and label-free approaches in the third chapter (Ab, reporter, Calcium Imaging). Maybe distinguish more between live-imaging and fixed specimens? If this could be fixed I would be happy.

Round 2

Reviewer 1 Report

The paper is accepted in this form.